# Mitofusin-2 Down-Regulation Predicts Progression of Non-Muscle Invasive Bladder Cancer

**DOI:** 10.3390/diagnostics11081500

**Published:** 2021-08-20

**Authors:** Antonella Cormio, Gian Maria Busetto, Clara Musicco, Francesca Sanguedolce, Beppe Calò, Marco Chirico, Ugo Giovanni Falagario, Giuseppe Carrieri, Claudia Piccoli, Luigi Cormio

**Affiliations:** 1Department of Biosciences, Biotechnologies, and Biofarmaceutical, University of Bari, 70126 Bari, Italy; antonella.cormio@uniba.it; 2Department of Urology and Renal Transplantation, University of Foggia, 71122 Foggia, Italy; chirico.marco90@gmail.com (M.C.); ugofalagario@gmail.com (U.G.F.); giuseppe.carrieri@unifg.it (G.C.); luigi.cormio@unifg.it (L.C.); 3CNR Institute of Biomembranes, Bioenergetics and Molecular Biotechnologies (IBIOM), 70126 Bari, Italy; c.musicco@ibiom.cnr.it; 4Department of Pathology, University of Foggia, 71122 Foggia, Italy; francesca.sanguedolce@unifg.it; 5Department of Urology, Bonomo Hospital, 76123 Andria, Italy; beppecalo49@gmail.com; 6Department of Clinical and Experimental Medicine, University of Foggia, 71122 Foggia, Italy; claudia.piccoli@unifg.it

**Keywords:** bladder cancer, NMIBC, Mfn2, ClpP, disease progression, disease recurrence

## Abstract

Identification of markers predicting disease outcome is a major clinical issue for non-muscle invasive bladder cancer (NMIBC). The present study aimed to determine the role of the mitochondrial proteins Mitofusin-2 (Mfn2) and caseinolytic protease P (ClpP) in predicting the outcome of NMIBC. The study population consisted of patients scheduled for transurethral resection of bladder tumor upon the clinical diagnosis of bladder cancer (BC). Samples of the main bladder tumor and healthy-looking bladder wall from patients classified as NMIBC were tested for Mfn2 and ClpP. The expression levels of these proteins were correlated to disease recurrence, progression. Mfn2 and ClpP expression levels were significantly higher in lesional than in non-lesional tissue. Low-risk NMIBC had significantly higher Mfn2 expression levels and significantly lower ClpP expression levels than high-risk NMIBC; there were no differences in non-lesional levels of the two proteins. Lesional Mfn2 expression levels were significantly lower in patients who progressed whereas ClpP levels had no impact on any survival outcome. Multivariable analysis adjusting for the EORTC scores showed that Mfn2 downregulation was significantly associated with disease progression. In conclusion, Mfn2 and ClpP proteins were found to be overexpressed in BC as compared to non-lesional bladder tissue and Mfn2 expression predicted disease progression.

## 1. Introduction

Bladder cancer (BC) is the most common urinary system malignancy. The reported European age-standardized incidence rate is 16.3 [1]. Non-muscle invasive BC (NMIBC) accounts for nearly 75% of newly diagnosed BC cases whereas the remaining cases present as muscle-invasive BC (MIBC); over 50% of NMIBCs recur, while 15–20% advance towards a muscle-invasive form [2]. BC development and progression is a complex multistep process whereby several molecular pathways seem to be involved [3].

Dysregulation of the energy metabolism has emerged as a hallmark of many cancers [4]. One of the best known modified cellular metabolism found in cancer cells is known as the Warburg effect [5]. In this condition, cancer cells actively metabolize glucose and produce an excess of lactate even in the presence of oxygen. Glycolysis has been therefore considered as the major metabolic process for energy production and anabolic growth in cancer cells. However, the past decade has witnessed an explosion of knowledge regarding how mitochondria may play a key role in cancer cells’ energy production, by oxidative phosphorylation, and may influence cancer initiation and progression [4,6,7].

As for BC, available evidence suggests that the main energy source sustaining proliferation may be glycolysis [8,9], thus indicating that mitochondrial dysfunctions may play a role in carcinogenesis. Indeed, mitochondrial DNA (mtDNA) exhibits a high rate of mutations in both human and rat BC [10,11,12]. Therefore, the role of mitochondrial function in BC seems to be worth investigating [11].

Recent reports highlight the role of mitochondrial proteins Mitofusin-2 (Mfn2) and caseinolytic protease P (ClpP) in cancer progression. Mfn2 is a mitochondrial protein involved in mitochondrial dynamics (fusion) as well as in the cell cycle [13], cell proliferation [14], and apoptosis [15], somehow balancing mitochondrial function [16]. Mfn2 is downregulated in many human tumors [17,18,19,20,21,22] including BC and is correlated with shortened survival. Recent studies indicated that Mfn2 has a key role in cancer progression; however, whether it acts as a tumors suppressor or oncogene in this process remains controversial. In fact, Mfn2 is downregulated in many human tumors including BC [17,18,19,20,21,22]. However, other studies reported Mfn2 overexpression in lung and gastric cancer cells compared to non-neoplastic tissues [23,24].

ClpP is a mitochondrial serine protease that plays an important role in maintaining the integrity of the mitochondrial respiratory chain and optimizing mitochondrial function by removing damaged mitochondrial proteins [25]. ClpP is upregulated in many primary and metastatic human tumors [19,26,27,28,29] and in some tumors is correlated with shortened survival [26]. Its overexpression is also associated with mitochondrial stress [28]. Chemical or genetic inhibition of ClpP impairs oxidative phosphorylation (OXPHOS) and mitochondrial functionality leading to an anti-proliferative effect in different cell types relying on OXPHOS [26,27,28,29,30]. To date, overexpression of ClpP has been reported in BC tissue [26,29] but its potential prognostic and predictive role in BC is still to be determined.

The present study, therefore, aimed to determine the prognostic role of Mfn2 and ClpP proteins expression in patients diagnosed with BC.

## 2. Materials and Methods

### 2.1. Patients 

The study population consisted of patients with non-muscle invasive BC at first diagnosis scheduled for transurethral resection of bladder tumor (TURBT) at Foggia University Hospital, Italy. A total of 66 patients with NMIBC met the inclusion criteria; their median age was 72 years and the female gender accounted for 20% of cases (Table 1). Patients who had previously undergone radiotherapy or systemic chemotherapy for any reason and patients in whom the study procedure could have impaired final pathology were excluded from this study.

A single senior pathologist with expertise in bladder pathology reviewed all specimens according to the latest WHO Classification of Tumors of the Urinary System and Male Genital Organs [31] and the 8th Edition of the AJCC-TNM classification of BC [32]. In patients undergoing TURBT, samples of the main bladder tumor and healthy-looking bladder wall (control tissue) were obtained by a cold-cut biopsy to avoid thermal modifications of the samples; moreover, great care was used to sample neoplastic and healthy-looking tissues to avoid contamination of non-cancer cells and non-urothelial cells, respectively. All samples were snap-frozen in liquid nitrogen and stored at −80 °C.

Patients were followed up according to EAU guidelines for NMIBC [2]. The low-risk group were those with a primary, single, Ta/T1 low-grade tumor, <3 cm in diameter and without CIS. All other tumors were classified as high-risk. Tumor recurrence was defined as pathological evidence of disease at bladder biopsy or TURBT, whereas tumor progression was defined as a pathological shift to MIBC or shift to metastatic disease (M+).

The study protocol was approved by the Ethical Committee of the University of Foggia (Decision n. 6/CE/2019 of 9 January 2019; Ethical Committee at the University Hospital “Ospedali Riuniti”, Foggia, Italy) and carried out following the Helsinki Declaration recommendations. All patients signed an informed consent to be enrolled.

### 2.2. Mfn2 and ClpP Expression Level 

Assessment of Mfn2 and ClpP was carried out using Western-blotting analysis as previously reported [19]. The samples (20–50 mg) were homogenized in lysis buffer (NaCl 150 mM, Hepes 50 mM pH 7.9, Triton X100 0.5%) and homogenates were incubated on ice for 30 min and clarified by centrifugation at 10,000× *g* for 10 min. The mouse anti-Mfn2 monoclonal antibody (1:10,000, Abnova Corporation, Taipei, Taiwan) and the rabbit anti-ClpP monoclonal antibody (1:50,000, Abcam, Cambridge, UK) were used. Mfn2 and ClpP levels were referred to Actin content (loading control).

### 2.3. Statistical Analysis

Continuous variables were reported as medians and interquartile range (IQR) and analyzed by the Mann-Whitney test. Categorical variables were reported as frequencies and compared by the Chi-Square or the Fisher’s exact test, as appropriate. 

The expression levels of Mfn2 and ClpP were first compared in lesional and non-lesional tissue, and then across low-risk NIMBC, including low-grade Ta BC, and high-risk NMIBC, including all T1 tumors, high-grade Ta/T1 tumors, and carcinoma in situ (CIS) [32]. The expression levels of these proteins were then correlated to disease recurrence, progression, and cancer-related death. 

Since Mfn2 was found to be a predictor of disease progression and to establish optimal cutoffs for the biomarker of interest, we performed receiver operating characteristic (ROC) analysis predicting disease progression at 24 months of follow-up. Youden’s index was then used in conjunction with ROC analysis to select the cutoffs of the mitochondrial proteins associated with the best combination of sensitivity and specificity. The index was defined for all points of the ROC curve, and the maximum value of the index was used as a criterion for selecting the optimum cut-off point. The sensitivity and specificity of the mitochondrial proteins were computed and graphically presented at each probability cutoff. 

Recurrence-free survival (RFS), progression-free survival (PFS), and cancer-specific survival (CSS) of patients with proteins downregulation (below the cut-off) and upregulation (above the cut-off) were estimated non-parametrically using the Kaplan-Meier method, with differences being tested for significance using the Log-rank test. 

Finally, the ability of these mitochondrial proteins in predicting the recurrence and progression of NMIBCs was tested by Cox regression analysis. Univariable semi-parametric Cox regression analysis was used to match them with the variables included in EAU guidelines [32], namely gender, tumor size (>3 cm), primary vs. recurrent tumor, multifocality, presence of CIS, tumor grade and stage, and EORTC score. Then, multivariable models were developed to test the added value of these two novel biomarkers to the EAU-validated parameters for risk assessment.

Significance was set at *p* < 0.05 and all tests were two-sided. Statistical analysis was carried out using the STATA SE 14 (Stata Corp, College Station, TX, USA) according to the following syntax: chi2, kwallis, lsens, stset, sts test, stcox.

## 3. Results

### 3.1. Mfn2 and ClpP Proteins Were Overexpressed in BC as Compared to Non-Lesional Bladder Tissue

Overall, median Mfn2 expression levels were significantly higher (2.6 vs. 0.5; *p* < 0.0001) in lesional than in non-lesional tissue; such difference was even higher (1.1 vs. 0.1; *p* < 0.0001) for ClpP expression levels (Table 1).

Lesional Mfn2 expression levels were significantly higher in low-risk NMIBC as compared to high-risk NMIBC (*p* = 0.0005) whereas non-lesional levels were similar; conversely, lesional ClpP expression levels were significantly higher (*p* = 0.046) in high-risk NMIBC as compared to low-risk NMIBC; again, non-lesional levels were similar in the 2 categories (Table 1). Representative western blotting of Mfn2 and ClpP in bladder tissues is presented in Figure 1.

Median follow-up was 33.0 (IQR: 30.0, 39.0) months (Table 1). Among the 66 total patients, disease recurrence occurred in 23 (34.8%), progression in 8 (12.1%), and cancer-related death in 5 (7.6%). Those patients who experienced recurrence had lower lesional Mfn2 expression levels but the difference was not statistically significant (3.2 vs. 2.4; *p* = 0.3), whereas those who experienced progression had significantly lower lesional Mfn2 expression levels (3.2 vs. 1.2; *p* = 0.024). Mfn2 expression levels were also slightly lower in NMIBC patients who died of BC (3.2 vs. 1.5; *p* = 0.3).

Lesional ClpP expression levels were slightly lower in patients who recurred (1.2 vs. 1.1; *p* = 0.6), progressed (1.1 vs. 0.9; *p* = 0.3) and died of BC (1.2 vs. 0.5; *p* = 0.2) (Table 2).

### 3.2. Prognostic Value of Mfn2 and ClpP Down-Regulation in Patients with NMIBC

Sensitivity and specificity of the mitochondrial proteins in the prediction of disease progression at 24 months of follow-up are presented in Appendix A. The cutoffs of Mfn2 and ClpP with the highest Youden Index and thus, with the best combination of sensitivity and specificity were 1.75 and 1.5 for Mfn-2 and ClpP, respectively. Kaplan-Meier curves showed that Mfn2 downregulation had a significantly worse PFS (*p* = 0.0008) but did not impact on RFS and CSS (Figure 2). ClpP down-regulation had no impact on any survival outcome (Figure 2).

Univariable Cox regression analysis (Table 3) pointed out that T1 stage was the only significant predictor of disease recurrence (HR: 2.60; *p*-value: 0.042) whereas presence of CIS (HR: 13.00; *p*-value: 0.001), EORTC score (HR: 1.19; *p*-value: 0.013) and Mfn2 down-regulation (HR: 9.54; *p*-value: 0.006) all emerged as significant predictors of disease progression. At multivariable analysis adjusting for EORTC scores, however, Mfn2 downregulation was the only significant predictor of disease progression (HR: 8.78; *p*-value: 0.009).

## 4. Discussion

The increasing interest in mitochondrial metabolism changes has been supported by the evidence of their involvement in cancer cell proliferation [4,5,6,7,8,9] and by their potential role as novel therapeutic targets. Oresta et al. investigated the mechanism by which mitomycin C (MMC) reduces recurrences on NMIBC, revealing that MMC favors immunogenic cell death and tumor protection through metabolic reprogramming of tumor cells towards increased OXPHOS and increasing mitochondrial permeability. Such changes lead to the cytoplasmic release of mitochondrial DNA, thus activating inflammation with interleukin-1β secretion which ultimately promotes dendritic cell maturation [33]. Resistance to immunogenic cell death (ICD) and higher recurrence incidence after chemotherapy in patients with NMIBC was associated with mitochondrial dysfunction related to low abundance of complex I of the respiratory chain. 

The mechanisms underlying the transition from mitochondrial OXPHOS to aerobic glycolysis in cancer cells are not yet fully understood. Mitochondrial dynamics (fusion and fission) may play a role in this transition. Indeed Mfn2, a key mediator of mitochondrial fusion, supports increased oxidative phosphorylation [34]. 

The first study testing Mfn2 expression in BC pointed out that tumor compared to adjacent non-tumorous tissue was characterized by decreased Mfn2 protein levels and significantly higher Mfn2 mRNA levels [18]. No correlation was found between Mfn2 expression and clinical features, however, the authors pointed out that Mfn2 overexpression in BC cells significantly inhibited cell proliferation and induced apoptosis, thus suggesting that Mfn2 may act as a tumor suppressor gene and can represent a potential therapeutic target [18]. 

A recent study [17] reported a decrease of Mfn2 mRNA levels in BC compared to normal tissues and an association between low Mfn2 expression levels and high tumor grade, stage, lymph node involvement, and shorter overall survival. The authors did not compare Mfn2 expression in NMIBC and MIBC samples grouping together T1 and T2 patients versus T3 and T4 patients. Additionally, they did not test the prognostic role of Mfn2 for disease progression and recurrence. On the other hand, this study pointed out that Mfn2 silencing in BC cells promoted proliferation, migration, and invasion in vitro models and enhanced tumor progression in vivo suggesting that BC progression may be delayed by increasing Mfn2 expression [17].

ClpP has been found to be increased in several cancers including BC [19,26,27,28,29] and its expression has been found to be associated with poor prognosis [26,27]. Moreover, ClpP silencing has been shown to inhibit proliferation, migration, invasion, as well as to promote cell apoptosis [26,27,28,29]. Nevertheless, to the best of our knowledge, the correlation between ClpP expression and clinical outcome has never been studied in BC.

Our study pointed out that BC displayed significantly higher expression levels of Mfn2 and ClpP than healthy bladder tissue. Interestingly, lesional Mfn2 expression levels significantly decreased by increasing disease stage/grade whereas non-lesional levels remained stable; conversely, lesional ClpP expression levels were significantly higher (*p* = 0.046) in high-risk NMIBC as compared to low-risk NMIBC. Again, non-lesional levels remained stable through the 2 categories. 

Our data, in contrast with Pang et al. 2019 [17], showed an increased Mfn2 expression level in BC tissues compared to adjacent non-tumor tissues, however, similarly to these authors we found a reduction in Mfn2 expression levels in high-grade BC tumors. We hypothesize that the first phase of BC carcinogenesis is characterized by high levels of Mfn2, which however tends to decrease by increasing disease stage and grade, thus making Mfn2 down-regulation a marker of aggressiveness and predisposition to progress in patients with NMIBC. Indeed, cox regression analysis demonstrated that Mfn2 down-regulation was associated with significantly worse PFS. Most important, the EORTC score-adjusted multivariable analysis demonstrated that, in patients with NMIBC, Mfn2 down-regulation was the only significant predictor of disease progression. 

It has been reported that mitochondrial fusion, along with an increase of stress-induced mitochondrial protease, protects cells from apoptosis and induces a metabolic shift from glycolysis to oxidative phosphorylation [34,35,36,37].

Therefore, it is attractive to assume that the increase of Mfn2 and ClpP expression in lesional compared to non-lesional tissues may increase oxidative phosphorylation and promote energy production and proliferation in the first phase of carcinogenesis. On the contrary, the Mfn2 down-regulation in high-risk NMIBC and in patients who progress may be explained assuming a shift from oxidative phosphorylation to glycolysis that is a more favorable condition in a future hypoxic microenvironment.

Supporting this hypothesis, previous studies have shown that HIF-1α significantly increases in high-grade, invasive, and metastatic bladder cancer tissues compared with low-grade tumors [38]. Therefore, it can be envisioned that a low level of Mfn2 in NMIBC may predispose to progression and a worse prognosis.

Conversely, despite its role in other cancers, ClpP expression was not found to be associated with BC outcomes.

Given the unpredictable behavior of NMIBC, particularly in the high-risk group, the identification of novel molecular markers predicting disease outcome and representing potential therapeutic targets remains a major clinical issue. Our findings would suggest that Mfn2 expression may play such a role. 

It is worth mentioning that targeting cancer cells metabolism could be particularly relevant given the increasing interest in immune therapies. Indeed, the interplay between tumor and immune cells leads to metabolic competition in the tumor ecosystem, limiting nutrient availability and leading to microenvironmental acidosis, which hinders immune cell function [39,40].

The novelty of this study was to evaluate the role of Mfn2 and ClpP as molecular predictive markers in a specific set of patients, namely NMIBC. However, this study has some limitations. First, the study population is relatively small but we elected to design a prospective study focusing on a homogeneous population of NMIBCs with a reasonable follow-up. Second, it did not include in vitro functional tests onto cell lines, but this would have been outside the scope of the present study aiming to test the role of these markers as potential predictors of disease outcome. Larger multicenter studies should be performed to externally validate our findings.

In conclusion, in patients with NMIBC, the neoplastic tissue displays significantly higher Mfn2 and ClpP expression levels than the non-neoplastic tissue. Mfn2 expression correlates with disease outcome since down-regulation predicts disease progression, whereas ClpP expression does not. These findings make Mfn2 an interesting marker and a potential therapeutic target.

## Figures and Tables

**Figure 1 diagnostics-11-01500-f001:**
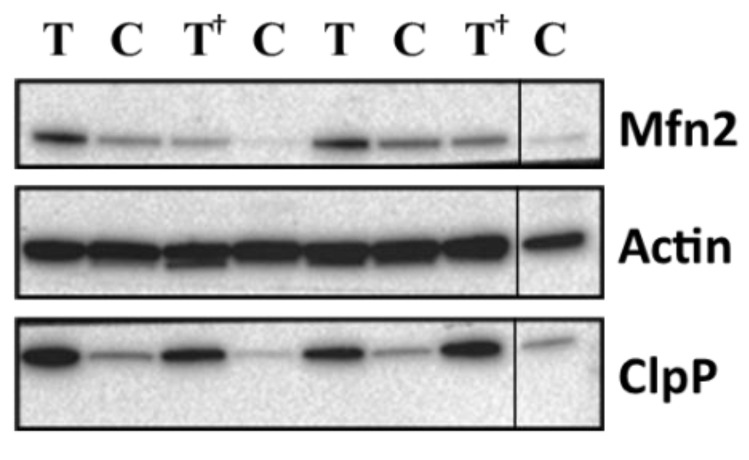
Representative western blotting of Mfn2 and ClpP proteins in bladder tissues. Proteins levels are referred to Actin content (loading control). T, lesional tissue (Tumor); C, adjacent non-lesional tissue (Control). ^†^, high-risk NMIBC patients. The grouping of blots is cropped from different parts of the same gel.

**Figure 2 diagnostics-11-01500-f002:**
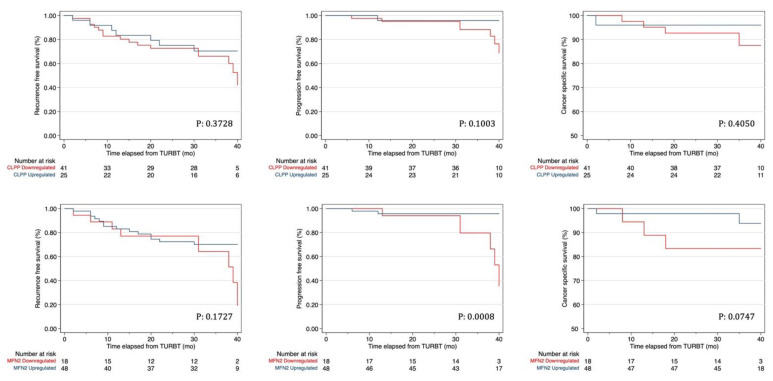
Kaplan–Mayer curves for recurrence, progression, and cancer-specific survival in patients with NMIBC according to down- and upregulation of Mfn2 and ClpP.

**Table 1 diagnostics-11-01500-t001:** Patients’ characteristics.

	Overall *N* = 66	Low-Risk NMIBC (*N* = 32)	High-Risk NMIBC (*N* = 34)	*p*-Value
**Male Gender, *n* (%)**	53 (80.0%)	23 (71.9%)	30 (88.2%)	0.13
**Primary tumor, *n* (%)**	35 (53.0%)	19 (59.4%)	16 (47.1%)	0.048
**Multifocal tumor, *n* (%)**	35 (53.0%)	22 (68.7%)	13 (38.2%)	0.014
**>3 cm, *n* (%)**	11 (16.6%)	2 (6.3%)	9 (26.5%)	0.048
**Concomitant Cis, *n* (%)**	3 (4.5%)	0 (0.0%)	3 (8.8%)	0.085
**Mfn2 lesional**	2.6 (0.8, 7.1)	4.6 (2.7, 9.0)	2.5 (1.2, 7.0)	**0.0005**
**Mfn2 non-lesional**	0.5 (0.3, 1.0)	0.5 (0.4, 1.3)	0.6 (0.3, 1.0)	0.7
**ClpP lesional**	1.1 (0.6, 1.6)	1.0 (0.6, 1.8)	1.3 (0.8, 3.8)	**0.046**
**ClpP non-lesional**	0.1 (0.1, 0.4)	0.1 (0.1, 0.3)	0.1 (0.1, 0.3)	0.4
**Follow-up months**	33.0 (30.0, 39.0)	36.5 (31.0, 41.0)	31.0 (30.0, 39.0)	
**Recurrence, *n* (%)**	23 (34.8%)	10 (31.2%)	13 (38.2%)	0.6
**Progression, *n* (%)**	8 (12.1%)	3 (9.4%)	5 (14.7%)	0.5

Continuous variables were reported as medians and interquartile range (IQR). In bold significant values.

**Table 2 diagnostics-11-01500-t002:** Lesional Mfn2 and ClpP expression levels according to disease recurrence, disease progression, and cancer-specific death.

	No Recurrence (*N* = 43)	Recurrence (*N* = 23)	*p*-Value
Mfn2	3.2 (2.0, 9.0)	2.4 (1.2, 8.3)	0.3
ClpP	1.2 (0.8, 2.5)	1.1 (0.5, 3.3)	0.6
	**No Progression (** ***N* = 58)**	**Progression (** ***N* = 8)**	***p*** **-Value**
Mfn2	3.2 (2.0, 9.0)	1.2 (0.4, 4.7)	**0.024**
ClpP	1.1 (0.8, 3.0)	0.9 (0.3, 1.4)	0.3
	**Alive (** ***N* = 61)**	**Cancer-Specific Death (** ***N* = 5)**	
Mfn2	3.2 (1.9, 8.5)	1.5 (1.3, 2.8)	0.3
ClpP	1.2 (0.8, 3.0)	0.5 (0.3, 1.1)	0.2

Continuous variables were reported as medians and interquartile range (IQR). In bold significant values.

**Table 3 diagnostics-11-01500-t003:** Univariable and multivariable Cox regression analysis evaluating predictors of Recurrence and Progression in patients with NMIBC. EORTC scores for Recurrence and Progression were used accordingly.

	Recurrence			Progression		
Covariate	H.R.	95% CI	*p* > |z|	H.R.	95% CI	*p* > |z|
**Univariable**						
Sex	2.18	0.88, 5.41	0.091	0.44	0.05, 3.63	0.449
Multifocal (vs. Single)	1.66	0.73, 3.81	0.228	0.79	0.19, 3.34	0.754
>3 cm (vs. <3 cm)	0.62	0.14, 2.68	0.524	1.56	0.17, 14.08	0.691
Recurrent (vs. Primary)	0.60	0.26, 1.39	0.235	0.35	0.07, 1.74	0.198
T1 (vs. Ta)	2.60	1.04, 6.54	**0.042**	3.47	0.80, 15.04	0.096
CIS, present (vs. absent)	2.48	0.73, 8.43	0.145	13.00	2.88, 58.58	**0.001**
G3 (vs. G1–2)	1.45	0.64, 3.30	0.373	2.57	0.61, 10.86	0.200
EORTC score, per unit	1.15	0.99, 1.35	0.069	1.19	1.04, 1.37	**0.013**
Mfn2 Downregulation	1.77	0.77, 4.11	0.181	9.54	1.90, 47.83	**0.006**
ClpP Downregulation	1.49	0.61, 3.64	0.379	4.89	0.60, 39.88	0.139
**Multivariable**						
Mfn2 Downregulation	1.51	0.64, 3.58	0.350	8.78	1.70, 45.31	**0.009**
EORTC score, per unit	1.14	0.97, 1.33	0.115	1.07	0.83, 1.39	0.578
**Multivariable**						
ClpP Downregulation	1.33	0.54, 3.29	0.538	4.31	0.52, 35.85	0.176
EORTC score, per unit	1.14	0.98, 1.33	0.091	1.12	0.86, 1.45	0.402

In bold significant values.

## Data Availability

Not applicant.

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
