# Peer review of "Mitofusin-2 Down-Regulation Predicts Progression of Non-Muscle Invasive Bladder Cancer"

_diagnostics, 2021, doi:10.3390/diagnostics11081500_

Round 1
Reviewer 1 Report
The authors answered the reviewers' comments accordingly. I have no further comments.
Author Response
Thank you for your comments.
Reviewer 2 Report
Regarding my concerns, the authors gave reasonable, but not complete, responses to my comments. If I am an author, I will perform ROC analysis to determine the cut-offs of Mfn2 and ClpP for downregulation.
There are some sentences which I cannot understand well. Please review the text carefully. English should be polished. There are two Table 2.
Author Response
We thank the reviewer for this comment. As suggested we added ROC analysis for MFN2 and CLPP. In order to select the optimal cut-off for the mitochondrial proteins, sensitivity and specificity in the prediction of the outcome of interest were computed and the cutoff was selected based on the highest Youden Index. A supplementary table 1 was added. Additionally, we apologize for the typos regarding Table 2 (corrected to table 3) and the text underwent a careful English revision. We thank the reviewer for his/her valuable and constructive comments.
This manuscript is a resubmission of an earlier submission. The following is a list of the peer review reports and author responses from that submission.
Round 1
Reviewer 1 Report
- How can the authors make sure that adjacent non-lesion tissue is actual non-lesion?
- Due to heterogeneity, it is not appropriate to use western immunoblotting to measure the expression of the proteins of interest for clinical correlation studies.
- For the univariate studies of recurrence, only T1 stage is significant. How can the authors use Mfn2 or ClpP in the univariate analyses?
- For the univariate analyses of progression, the authors may include CIS, Mfn2, and ETROC but not only Mfn2 and ETROC.
- The authors may carry out some experiments on molecular mechanisms.
Reviewer 2 Report
The manuscript entitled “Mitofusin-2 down-regulation predicts progression of non-muscle invasive bladder cancer” describes the potential of MFN2 and CLPP as a prognostic biomarker in patients treated with NMIBC. The manuscript is well-written and describes a well-designed, novel, and interesting study. However, the authors need to address a few comments and revise the manuscript accordingly.
Major points:
- To maximize the potential clinical impact of MFN2 and CLPP levels as outcome biomarkers, the Authors should do a receiver operating characteristic (ROC) curves analysis. This analysis allows to assess the MFN2 and CLPP accuracy as outcome biomarkers (AUC, sensitivity, specificity, positive and negative predictive values) and determine the best cutoff value of MFN2 and CLPP. In survival curves, patients should be dichotomized according to these cutoff values obtained from the ROC curves constructed to predict each event (death, cancer-specific death, progression, or recurrence).
- Why was the Kruskal Wallis test used to compare the means when only two groups are being compared? The correct test to compare two groups is the Mann–Whitney U test (if the distribution is non-normal).
- Table 1 should be improved and divide into two tables or one table and one figure. Table 1 should only include demographic and clinical features. This Table should include gender, age, primary tumor, tumor stage, tumor grade, tumor size, tumor location, recurrence, progression, death, follow-up, and other features used in the Cox regression. The expression level of MFN2 and CLPP should be in another Table or Figure.
Minor points:
- The inclusion criteria should be described in the Material and Methods section.
- Ta should be written out in full.
Reviewer 3 Report
Antonella et al. investigated expressions of mitofusin-2 (Mfn2) and caseinolytic protease P (ClpP) in 66 cases of non-muscle invasive bladder cancer (NMIBC) by Western blotting. They found that Mfn2 and ClpP were overexpressed in NMIBC and concluded that Mfn2 downregulation was associated with disease progression. It is interesting to focus on the role of glycolysis in the development and progression of bladder cancer. To focus on Mfn2 and ClpP is also new. However, I wonder whether the presented data could lead to their conclusion. I raise my concerns below.
1. The authors claim that Mfn2 functions as a tumor suppressor. However, Mfn2 was upregulated both in low-risk and high-risk NMIBC as compared with the corresponding normal mucosa, and this result contradicts their assumption. In addition, the authors described in the Discussion that the first phase of BC carcinogenesis is characterized by high levels of Mfn2 which however tend to decrease by increasing stage and grade, in agreement with Peng et al [17]. However, the cited reference does not suggest such a theory. They must explain why tumor suppressor Mfn2 is upregulated in low-risk NMIBC. If possible, I recommend analyzing Mfn2 expression in normal mucosa, low-grade NMIBC, and high-grade NMIBC side by side which were dissected from the same bladder at the same time to validate sequential change of Mfn2 expression level.
2. The cut-offs for downregulation of Mfn2 and ClaP were arbitrary decided. According to these levels, statistically significant effect on prognosis was elicited. However, I think that a rationale to decide a cut-off would be necessary.
3. This study included relatively a small number of NMIBC cases (n = 66). Without mechanical experiments, confirmation by using a validation cohort would be necessary.
4. Definition of low-risk and high-risk NMIBC had better be described in the text.